# Placing a Well-Designed Vegan Diet for Slovenes

**DOI:** 10.3390/nu13124545

**Published:** 2021-12-18

**Authors:** Boštjan Jakše

**Affiliations:** Department of Food Science, Biotechnical Faculty, University of Ljubljana, SI-1000 Ljubljana, Slovenia; bj7899@student.uni-lj.si

**Keywords:** vegan diet, plant-based, health, sport, environment, barriers, education, coronavirus

## Abstract

Interest in vegan diets has increased globally as well as in Slovenia. The quantity of new scientific data requires a thorough synthesis of new findings and considerations about the current reserved position of the vegan diet in Slovenia. There is frequently confusion about the benefits of vegetarian diets that are often uncritically passed on to vegan diets and vice versa. This narrative review aims to serve as a framework for a well-designed vegan diet. We present advice on how to maximize the benefits and minimize the risks associated with the vegan diet and lifestyle. We highlight the proper terminology, present the health effects of a vegan diet and emphasize the nutrients of concern. In addition, we provide guidance for implementing a well-designed vegan diet in daily life. We conducted a PubMed search, up to November 2021, for studies on key nutrients (proteins, vitamin B_12_, vitamin D, omega-3 long chain polyunsaturated fatty acids (eicosapentaenoic acid (EPA) and docosahexaenoic acid (DHA)), calcium, iron, zinc, iodine and selenium) in vegan diets. Given the limited amount of scientific evidence, we focus primarily on the general adult population. A well-designed vegan diet that includes a wide variety of plant foods and supplementation of vitamin B_12_, vitamin D in the winter months and potentially EPA/DHA is safe and nutritionally adequate. It has the potential to maintain and/or to improve health. For physically active adult populations, athletes or individuals with fast-paced lifestyles, there is room for further appropriate supplementation of a conventional vegan diet according to individuals’ health status, needs and goals without compromising their health. A healthy vegan lifestyle, as included in government guidelines for a healthy lifestyle, includes regular physical activity, avoidance of smoking, restriction of alcohol and appropriate sleep hygiene.

## 1. Introduction

Contemporary dietary guidelines need to focus on health, the environment and animal welfare; however, they are currently threatening all three [1,2]. Dietary and lifestyle changes towards a healthy and sustainable diet/lifestyle could be associated with reductions in premature mortality from diet/lifestyle-related noncommunicable diseases, reductions in environmental impacts, COVID-19 severity and management and preventing the next coronavirus pandemic [3,4,5,6,7]. According to the United Nations, The American Public Health Association and several experts in the time of the COVID-19 pandemic, we should focus on dealing with the root causes of this epidemic spread, which include farming methods that crowd huge numbers of animals into small spaces [2,8].

There is increasing interest in eating vegan diets [9]. In a survey from 2018 across 28 countries, 3% of the global adult population was identified as vegan (e.g., researchers tested 20,313 adults, of which 575 were vegans) [10]. Currently, the worldwide prevalence of vegetarianism (which also includes people on a vegan diet) is not uniform [11]. In addition, generally, income is a significant driver for the type of diet people choose to follow (i.e., higher incomes are associated with diets rich in animal proteins rather than carbohydrate-based staples) [10]. Importantly, in the above-mentioned international survey, 20% of vegans have followed the diet only for about a year [10]. Regardless, different motives can lead to the adoption of a vegan diet. According to several studies, common motives for choosing a vegan diet include ethical and health-related benefits, body mass management, environmental concerns and religious reasons [12,13,14,15,16,17,18]. With that in mind, this has led to the development of new vegan products for the consumer market, including meat alternatives, other vegan alternatives [17] and plant-based protein supplements [19,20].

A study that used Google Trends to explore global popularity (the number of searches on vegan-related terms as a fraction of total searches in a given country, of various search categories) showed that the popularity of veganism was at an all-time high in 2020. Vegan-related queries were most popular in the United Kingdom, Australia, Israel, New Zealand and Austria, while Slovenia was in 15th place [21]. During the last six decades, the vegan diet has been rigorously investigated. During this time, 1440 articles were published by 4586 researchers. The leading country was the United States (471 articles), while the most prolific institutions were the University of Oxford (Oxford, UK; 59 articles) and Loma Linda University (Loma Linda, CA, United States, 38 articles) [22]. Existing data support the position that a shift towards a well-designed vegan diet is beneficial for sustainable food systems, planetary health and the prevention of common noncommunicable diseases [1]. However, although various professional associations have recognized a well-designed vegan diet as beneficial and healthy for people at all stages of life [23,24,25,26,27,28,29], others have expressed concerns about an inadequacy in terms of a lack of certain nutrients and its consequences [30,31].

Although there is ample evidence of the health benefits of a well-designed vegan diet, vegans in Slovenia are frequently faced with prejudice on unsubstantiated grounds [30,32,33,34,35,36,37]. Discussions about vegan diets are often emotionally charged, which is also due to a lack of education and professional knowledge or because of conflicts of interest and ignorance of the results of research on a vegan diet. Furthermore, there is considerable debate about a vegan diet and lifestyle, especially concerning nutritional deficiencies, health benefits, motives for adopting vegan diet and the difficulties in implementing a well-designed vegan diet (i.e., concerns that it is complicated, expensive and appropriate only for highly motivated individuals) and healthy lifestyle into everyday life.

Our narrative review of the literature summarizes the evidence on topics related to a well-designed vegan diet (Figure 1): (1) The definition of a vegan diet; (2) Health effects of a vegan diet; (3) Environmental footprint of a vegan diet; (4) Nutrients of concern in a vegan diet; (5) Practical recommendations for implementing a vegan diet; (6) Barriers to the implementation of a vegan diet. Importantly, this review of scientific literature is not meant to be a “recipe book” and does not provide sufficient detail for the immediate implementation of a well-designed vegan diet for a particular individual. Moreover, our aim is to analyse existing studies to incorporate information on individual topics that contributes to solid conclusions. We adopt this approach to the reviewing literature not because of the possible failure of support for a vegan diet using classic methodology but to provide to the Slovenian audience a comprehensive view of the current positioning of the vegan diet. It may serve as a springboard to establish appropriate planning that suits individuals’ dietary needs, health status, goals and personal circumstances.

## 2. The Definition of a Vegan Diet

The vegan diet (e.g., in terms of nutrition, not ideology) excludes all animal-based foods (i.e., meat, fish, dairy and eggs) but includes a wide variety of plant-based food groups (i.e., fruits, vegetables, grains, legumes, nuts and seeds, spices and herbs, mushrooms and algae) [38,39]. However, it does not require the consumption of whole foods or the restriction of fat or refined sugar [40]. Notably, in this review we maintain a distance from other subclassifications of vegan diets, namely, the raw vegan diet, the frugivorous diet and the macrobiotics diet [11].

For a vegan diet to provide documented health benefits, it must be well-designed weekly and daily from a variety of plant-based food groups that are eaten often enough, meals prepared with healthy methods of preparation (i.e., boiling, steaming, baking on baking paper) and eaten meals that induce satiety [41].

Importantly, there are many different definitions of a vegan (plant-based) diet, which makes comparisons among existing vegan diets and with other dietary patterns difficult. Consequently, this may cause significant confusion among researchers and the public [42].

In general, when referring to a vegan lifestyle, all the structures of recommendations for a healthy and active lifestyle for the general population should be considered, such as regular and appropriate physical activity (PA), nonsmoking, limited alcohol consumption, avoidance of prolonged sitting and appropriate sleep hygiene [43,44,45].

## 3. Health Effects of a Vegan Diet

### 3.1. In Adults

#### 3.1.1. Body Mass and Body Composition

In the modern world, weight and obesity are among the main global public health challenges [46,47,48]. More than fifty obesity-related diseases exist, which further increases the global burden of obesity [49]. Obesity is defined as the accumulation of excess body fat and not simply excess body mass; this is very important for individuals who are within the normal body mass index (BMI) class (e.g., those with so-called sarcopenic obesity) [50,51]. Sarcopenic obesity is currently a significant public health problem with increasing prevalence worldwide (e.g., up to 42% of adults) [52,53].

The most recent data in a Slovenian report (e.g., national cross-sectional food survey, SI. Menu, that collected food consumption data in the period from March 2017 to April 2018 [54]) showed that 22% of adolescents, 39% of adults and 43% of older adults were overweight, and a similar proportion was observed in the obese class (6% of adolescents, 20% of adults and 32% of older adults). These numbers are in line with the assessed average body fat percentage (BF %) (measured by bioimpedance) for adult and older females (33% and 37%) as well as for adult and older males (25% and 29%) [55].

Vegan diets have been explored with various study designs (i.e., randomized/nonrandomized (un)controlled, crossover, cross-sectional, a meta-analysis of intervention/observational studies and narrative review) and have shown reduced body mass [56,57,58,59,60,61,62,63,64,65,66,67] and improved body composition status [59,60,67,68,69,70,71,72,73]. For example, a 12-week randomized controlled trial (“the Broad study”) showed substantial body mass loss results in overweight or obese patients with type 2 diabetes or some form of cardiovascular disease within 3 months (−8.6 kg), 6 months (−12.1 kg), and 12 months (−11.5 kg) of follow-up without mandating regular PA or restricting total energy intake [56]. Furthermore, our intervention study of a community-based whole-food plant-based lifestyle program (i.e., a supplemented vegan diet, 45 min of moderate-intensity PA 2–3 times weekly, and a support system) also showed substantial body mass loss with effective muscle mass preservation [59,68]. Regular and appropriate PA is extremely important, as body fat loss and muscle mass preservation are crucial for improvements in body mass and body composition [51,74,75], better visual body image [76,77] and appetite control [78,79,80]. Physical activity, as an important lifestyle component and one of the key factors for body mass management, has been found to have an extremely beneficial effect on various aspects of health [81,82]; however, appropriate body mass/body composition is more effectively addressed by combining both dietary changes and PA [51,83,84].

Therefore, a vegan lifestyle in terms of effective and healthy body mass loss or successful control of proper body mass and body composition includes a well-designed vegan diet combined with a healthy and active lifestyle. This means including at least 150–300 min/week of vigorous-intensity PA (e.g., resistance workouts) or some equivalent combination of moderate-intensity and vigorous-intensity aerobic PA as recommended by the World Health Organization (WHO) guidelines for healthy adults. Furthermore, the WHO also recommends that an individual add two or more times a week a moderate-intensity to high-intensity resistance workout [85].

#### 3.1.2. Cardiovascular Health

Currently, cardiovascular diseases (CVDs) remain the leading global cause of death, accounting for as many as one-third of all deaths [86]. In 2016, almost 50% of CVD deaths in the European Region were a consequence of inadequate diet, characterized by a low intake of whole grains, nuts, seeds, fruits and omega-3 fatty acids but a high intake of sodium [87]. The most important risk factors are dyslipidaemia, arterial hypertension and diabetes type 2 [88,89]. In addition to an unhealthy diet, the most common environmental factors influencing the development of cardiovascular disease are smoking, excessive alcohol intake and lack of PA [89,90]. Therefore, due to its efficacy, safety and affordability, dietary change has been recognized as the first-line intervention to prevent elevated cholesterol levels and CVD (deaths) [87,91,92].

In the most comprehensive study in Slovenia in terms of CVD prevention, researchers tested 500,000 adults (25% of all Slovenians). The results were alarming: elevated cholesterol levels were found in 69% of examined adult individuals [93]. Furthermore, according to data from the National Institute of Public Health of Slovenia, almost 50% of adults had elevated blood pressure [94]. Moreover, the results of an epidemiological study on the prevalence of elevated blood pressure in Slovenia between 2007–2009 (e.g., researchers tested 3422 adults of the 15,000 invited) found elevated blood pressure in 64% of examined adult individuals [95].

Many study designs have shown that a vegan diet successfully reduces common CVD risk factors or is associated with beneficial short- and long-term health effects (i.e., low-density lipoprotein cholesterol, triglycerides, blood pressure) [56,60,61,65,67,68,71,96,97,98,99,100,101,102,103,104,105,106,107]. A meta-analysis of 31 observational studies comparing the effect of a vegan diet with an omnivorous diet on CVD risk factors showed that vegans’ average LDL cholesterol values were 2.36 mmol/L, their triglycerides were 1.1 mmol/L and their blood pressure was 118/77 mmHg [63]. Similarly favourable findings were shown in a systematic review and meta-analysis of 30 observational and 19 clinical trials comparing the effect of a vegetarian diet (i.e., vegan diet included) with an omnivorous diet on plasma lipids. Consumption of vegetarian diets, particularly a vegan diet, was associated with lower levels of plasma lipids [108].

In many cases, a well-designed vegan diet can stop or even reverse the progression of CVD endpoints [99,100,101,102]. Researchers performed randomized controlled trials with long-term follow-up (“the Lifestyle Heart Trial”) on 48 patients with moderate to severe coronary heart disease [99]. The interventional program consisted of a low-fat (10%) “vegetarian” diet (with 12.4 mg/day of dietary cholesterol; practically a “vegan” diet), cessation of smoking, stress management training and moderate exercise. After 1 year, 82% of patients in the experimental group showed a significant change towards the regression of severe coronary atherosclerosis without the use of lipid-lowering drugs [109]. After 5 years, an even greater regression of coronary atherosclerosis measured by percent diameter stenosis was found [99]. In another study of 198 patients (177 adherent to a low-fat vegan diet, 21 nonadherent) with multiple comorbidities (i.e., hyperlipidaemia, hypertension, angina, myocardial infarction, diabetes), a 0.6% recurrent event rate of major cardiac events was observed after an average of 3.7 years of follow-up. Improvements were confirmed with positron emission tomography (PET scan) on myocardial reperfusion and angiographic evidence of disease reversal [100].

Given the available data, a well-designed vegan diet with some modifications in diet and PA and mandatory supervision by a CVD specialist who is also an expert on the vegan diet may be a viable option for individuals who are interested in vegan diets. This might be especially true given that CVDs are the leading global cause of mortality and that no other diet has shown such a profound beneficial effect. Although increasingly favourable scientific data exist regarding the beneficial effects of a vegan diet and CVD health, further well-designed studies are warranted.

#### 3.1.3. Diabetes Type 2

The worldwide prevalence of diabetes type 2 is rapidly increasing. It was estimated that in 2019, 463 million individuals had diabetes type 2. The projection suggests that by 2045, this number will reach as many as 700 million patients with diabetes type 2 [110]. Furthermore, the prevalence of diabetes (type 1 and type 2) in Slovenia is defined by data on prescribed blood glucose-lowering drugs, and the 2017 report has shown that approximately 111,400 individuals take medicines to lower blood glucose (8% of the population), which is a 33% increase compared to 2008 [111]. Moreover, according to data from the National Institute of Public Health of Slovenia, the total number of patients with diabetes was estimated to be 137,000, while the number of identified and undiagnosed patients was estimated to be as high as 211,000 [112].

In a recent review of the literature, researchers support the feasibility of diabetes type 2 remission in diabetic patients using lifestyle as the primary means of treatment, whereas PA appears to have a minor role compared to the dietary regimen; however, it may significantly improve insulin resistance and beta-cell activity [113]. In addition, it has been demonstrated that with regard to diabetes type 2 outcomes, a well-designed vegan diet in short- and long-term settings has the greatest success compared to a controlled diet and other dietary interventions [67,114,115,116,117,118]. A vegan diet (*n* = 49) and a conventional “diabetes-type-diet” (*n* = 50) in the treatment of type 2 diabetes were tested in a long-term (74-week) randomized controlled trial, which showed that the vegan diet significantly improved diabetes type 2 status (i.e., glycaemia and plasma lipid concentration, body mass reduction) [114]. Importantly, a vegan diet addresses core pathophysiologic mechanisms, insulin resistance and diminished beta-cell function. For example, beta-cell function and fasting insulin sensitivity improvement using a vegan diet were recently investigated in a 16-week randomized controlled trial on 75 overweight adults (*n* = 38 on a vegan diet, *n* = 37 on a control diet). Researchers have demonstrated that beta-cell function and fasting insulin sensitivity, both core pathophysiologic mechanisms of diabetes type 2, could be significantly improved [118]. Possible mechanisms for improvement of beta-cell function and insulin sensitivity that are suggested include reduced visceral and subfascial fat in muscle tissue or changed fat distribution (by well-designed (i.e., high in carbohydrates) vegan diet and exercise), improved incretin secretion, reduced lipotoxicity, glucotoxicity, oxidative stress and inflammation [118].

Since diabetes type 2 is a lifestyle disease, the overall data suggest that a well-designed vegan diet and lifestyle changes should be the first-line treatment for newly diagnosed patients with diabetes type 2, and may be an extremely powerful adjunctive method in patients with a long history of diabetes type 2 [113,119,120,121]. Importantly, adherence to a low-fat vegan diet/lifestyle to manage diabetes was comparable to and often higher than adherence to a conventional diet/lifestyle in several studies [122].

#### 3.1.4. Other Health Benefits

Since the available financial support is not sufficient to thoroughly investigate a vegan diet, it is worth mentioning that the literature confirms other health benefits of vegan diets. Some are of greater scientific value (i.e., liver and kidney health, favourable changes in gut microbiota, prostate cancer, women’s menstrual, menopausal and postmenopausal health, short-term mood state, migraine severity and frequency) [67,72,123,124,125,126,127,128]; however, some have limited scientific value (i.e., case studies about chronic kidney disease, diabetes type 1/insulin-dependent diabetes mellitus, low-grade follicular lymphoma, lupus nephritis and less common dementia in a large-scale observational Adventist Health study [129,130,131,132,133,134,135,136]). Notably, these findings are sufficiently impressive to warrant further examination of these results in well-designed studies.

For example, in a 16-week randomized control trial, an intervention group (*n* = 122) followed a low-fat vegan diet, and in a subset of participants (*n* = 44), hepatocellular and intramyocellular lipids were quantified by proton magnetic resonance spectroscopy. The hepatocellular and intramyocellular lipid levels decreased by 34% and 10%, respectively, while none of these variables significantly changed in the control group with no dietary changes [67]. In line with these results, a recent prospective study on 26 patients affected by nonalcoholic fatty liver disease (NAFLD) who adhered to a vegan diet for six months showed improved liver enzymes (e.g., ALT, AST and GGT values), and normalization of liver function tests as a whole was observed in 77% of patients [131]. Furthermore, in a 1-year randomized controlled trial on a group of 93 patients with early-stage prostate cancer (PSA: 4–10 ng/mL and cancer Gleason scores less than 7), 44 were supplemented (e.g., with soy protein, selected vitamins and minerals, nutrient-fortified plant-based foods, eicosapentaenoic acid (EPA, C20:5n-3) and docosahexaenoic acid (DHA, C22:6n-3) from fish oil supplementation) with a low-fat vegan diet/lifestyle (including moderate-intensity aerobic activity, stress management and group support). Researchers found remarkably decreased serum PSA values (4 of the baseline average) and inhibited LNCaP (lymph node carcinoma of the prostate) cell growth by 70% [124]. To further clarify the association between vegetarian and vegan diets and the risk for various chronic diseases and mortality from cancer, researchers conducted a meta-analysis of observational studies (86 cross-sectional and 10 cohort prospective) and found that a vegan diet conferred a significantly reduced risk (−15%) of the incidence of all cancers [38].

As these are different health issues, the recommendation refers only to the fact that a healthy diet, which is undoubtedly a well-designed vegan diet (e.g., which can be customized and/or supplemented), may offer beneficial health support to individuals facing various health challenges. Accordingly, the cooperativeness of an individual patient with a specialist physician is extremely important; however, we suggest in this regard a specialist who knows the results of the research on vegan diets well enough and, even better, who knows how to properly supervise the patient on a well-designed vegan diet.

#### 3.1.5. Sport and Exercise Performance

Gladiators and philosophers in ancient times were aware that strict vegetarian (vegan) diets are compatible with physical and intellectual performances [137,138]. Currently, there is still a common belief in the general population that a vegan diet may be detrimental to endurance and muscle strength. Clinical and cross-sectional studies comparing vegan, lacto-ovo-vegetarian and omnivorous diets have shown comparable [18,139,140,141,142] or better [143] aerobic capacity, anaerobic performance or muscle strength on a vegan diet [18,139,140,141,142,143]. A recent narrative review of observational and interventional studies of young and competitive athletes on vegan diets showed that they are able to tolerate and sustain higher physical burdens for longer durations and to recover from physical stress more rapidly. This supports the assumption that a well-designed vegan diet is compatible with peak performance sports [137] and is associated with a good health status [137,144].

Physically active individuals and athletes on a vegan diet have often added a variety of dietary supplements to their basic diet. The prevailing speculation is that plant-protein supplements are inferior to animal-based protein supplements in building muscle and increasing strength. However, vegan protein supplements (soy in a meta-analysis of nine long-term studies (≥6 weeks), and mycoprotein, a single-cell fungal protein, in a short-term study (3 days)), compared to animal protein supplements, offer equivalent support for muscle mass gain and strength in healthy older adults and in untrained participants engaged in resistance exercise [145,146]. In support of this contention are the results of a recent meta-analysis showing that soy protein supplementation in men undergoing resistance exercise training led to gains in muscle mass and strength similar to those observed in men supplemented with whey protein or other animal proteins [146]. In addition, a 6-week case study in a world-class vegan male professional powerlifter reported body composition changes [147]. The results suggested that a vegan diet supplemented with vitamins B_12_ and D, EPA/DHA, creatine and vegan protein combined with a resistance workout may support the loss of body mass and BF % but preserve and even increase muscle mass [147]. Furthermore, for physically active people and athletes, there is room for further appropriate supplementation of a conventional vegan diet according to individuals’ health status, needs and goals without compromising their health (e.g., until proven otherwise), through the use of plant-based protein supplements or meal replacements, sport drinks and creatine [16,140,148,149].

Importantly, we are aware that many famous and successful vegan athletes exist among us [137,150]; unfortunately, to our knowledge, we have no solid scientific data on their actual diet, lifestyle and health status. This provides an important incentive for researchers to obtain more data through scientific methods. In addition, research on the long-term effects of a vegan diet on sport performance or exercise benefits is still very limited; therefore, the current, albeit encouraging, evidence should be interpreted with caution.

### 3.2. During Pregnancy, Breastfeeding, Infancy and Childhood

Over the last twenty years, many professional associations in the field of nutrition and pediatrics have published their position statements/guidelines on the suitability of a vegan diet during pregnancy, breastfeeding, infancy and childhood [151,152]. The majority of them stand on the position that a well-designed vegan diet is healthy and appropriate for these particularly sensitive periods [151]. However, all of them clearly emphasize that a vegan diet requires more careful planning and appropriate supplementation [151,152]. Furthermore, the position of the European Society for Paediatric Gastroenterology, Hepatology, and Nutrition (ESPGHAN) is that a well-designed vegan diet can support children’s normal growth and development but should be used under appropriate medical or dietetic supervision [153]. Moreover, when breastfeeding is not possible in the case of a dairy-free diet in the first year of life, the North American Society for Pediatric Gastroenterology, Hepatology, and Nutrition (NASPGHAN) highlights that properly formulated vegan formulas (e.g., soy or rice-based formulas) can be an alternative to cow’s milk-based formulas [154].

Nutrition during the perinatal period and early development has an important impact on health into adulthood (i.e., adult body mass, common chronic diseases, food preferences) [153,155,156]. The suitability of a vegan diet can be assessed through nutritional adequacy and the incidence of pediatric diseases as well as chronic morbidity later in life (e.g., CVD, cancer). Therefore, during these delicate stages, it is extremely important to offer (i) appropriate guidance on a well-designed vegan diet and (ii) medical supervision for nutrients of concern. Special consideration is required to obtain the recommended amounts of the following nutrients on a vegan diet: vitamin B_12_ and D, EPA/DHA and iodine. With well-designed planning, the adequacy of energy intake and other nutrients (i.e., protein, calcium, iron, zinc and selenium) can be achieved. Twenty years ago, a narrative review established that a well-designed vegan diet can be adequate for children at all ages [157]. However, we do not have many studies of quality due to difficulties in recruiting parents of vegan children because parents fear criticism from the medical profession [158].

In two recent publications, a narrative review and a prospective paper, the researchers established evidence that a well-designed vegan diet during pregnancy and lactation is nutritionally complete, safe and healthy for both the mother and offspring, emphasizing that it should follow all the criteria that define it as adequate [159,160], with some adjustments compared to the guidelines for the normal adult population. Although more evidence is needed, based on the current wide range of research results on this topic, we can conclude that a well-designed vegan diet has a protective effect against poor pregnancy outcomes (e.g., obesity, preeclampsia and various pediatric diseases) and preterm delivery, and can support lactation [159,160]. Awareness must be raised about complete dietary intake, especially for nutrients of concern and necessary dietary supplements, according to international guidelines [153,159,160].

Similarly, a well-designed vegan diet is suitable for subsequent periods of life [23,151,159]. To express an objective position on the appropriateness of feeding children a vegan diet, and given the shortcomings of high-quality studies on this stage of the lifecycle, it is necessary to examine this topic from a broader perspective [161]. In brief, childhood obesity has become a major global epidemic related to substantial social and health burdens worldwide [162], which would not be the case if a well-designed vegan diet or any other form of recognized balanced diets were followed (e.g., vegetarian, omnivore).

Furthermore, four recent cross-sectional studies investigated the nutritional adequacy of infants/children who followed a nonvegan and vegan diet [163,164,165,166]. In the United States, research was conducted on 9848 children (aged 1–6 years) whose parents and caregivers completed 24 h dietary recall interviews. The results of the children who followed a “normal” diet might be seen as concerning. In nearly all of the children (99%), fiber deficiency was detected, 87% were deficient in vitamin D, 69% in vitamin E, 58% in potassium and 17% in calcium. Furthermore, the average consumption of DHA was well below the European Food Safety Authority (EFSA) recommendation (e.g., 70–100 mg/day) in nearly all children (97–99%), while 99.8% of the children exceeded the recommended sodium intake [164]. In Germany, the energy and macronutrient intake and the anthropometrics of 430 children (127 vegetarians, 139 vegans and 164 omnivores) aged 1–3 years was examined. As no significant difference in dietary intake and anthropometrics between the study groups was found, the authors concluded that a vegan diet in early childhood is capable of providing the same amount of energy and macronutrients and therefore enables normal growth. The results of the study also revealed that the children on a vegan diet consumed 2 times and 2.3 times less added sugar than children on a vegetarian diet and an omnivorous diet and 22% and 46% more fiber, respectively [163]. In Finland, a small cross-sectional study of 40 children (10 vegetarians, 6 vegans and 24 omnivores) with an average age of 3.5 years was conducted to determine their micronutrient status. The study found that fat intake was similar and that the intake of saturated fatty acids and cholesterol were significantly lower, whereas the intake of monounsaturated fatty acids, linoleic acid (LA, C18:2, n-6) and alpha linoleic acid (ALA, C18:3, n-3) were higher among children who followed a vegan diet than among those who consumed an omnivore diet. In addition, when comparing the nutrients of concern, children on a vegan diet compared with children on an omnivore diet consumed 93% more fiber, 277% more folate, 61% more iron and 14% more zinc (all without supplementation), and a similar intake of vitamin D (without supplementation) and iodine (with supplementation, e.g., iodized salt), but no EPA/DHA intake (without supplementation) [165]. Finally, the most recent cross-sectional study on 187 Polish children (63 vegetarian, 52 vegan and 72 omnivores) aged 5–10 years revealed an increased risk for B_12_ deficiencies in children who followed a vegan diet but not in those who were supplemented with vitamin B_12_ and vitamin D. In addition, the prevalence of possible B_12_ deficiency was 16%, 19% and 40% in omnivores, vegetarians and vegans, respectively. However, moderate and mild iron deficiency anemia was found among vegans and vegetarians in 2% (in both) and 6% and 7% of the sample (no case on omnivores) without severe iron deficiency anemia. Moreover, the prevalence of depleted iron stores was 13% in omnivores, 18% in vegetarians and 30% in vegans [166]. Given that vegan and vegetarian children in this study were recruited via advertising (i.e., web sites and social media that were targeting issues concerned with vegetarianism and veganism), their diets were presumably not optimally supervised by a nutrition expert. In this way, they could be given guidance on a well-designed vegan diet for their age group, beyond the advice on supplementation. Moreover, this information is of the utmost importance since the above-mentioned Finnish study showed that children on a vegan diet had an intake of 61% more iron compared with children on an omnivorous diet, while both groups of children had comparable iron stores (16 µg/L vs. 14 µg/L) [165].

It is well accepted that predispositions for CVD appear in childhood [167,168,169] and in pregnant women who are overweight or obese and/or have high blood pressure [170,171]. In Cleveland, a 4-week prospective randomized trial was performed on 30 pairs of obese children with hypercholesterolemia (aged 9–18 years) and their parents. They were randomized into two groups, one following a vegan diet without added fat and the other following the American Heart Association (AHA) diet. Both groups received 2 h weekly classes on nutritional education. While both diets demonstrated beneficial effects, in the vegan diet group, the beneficial changes were over 2 times more evident with regard to body mass/fat, systolic blood pressure, total and LDL cholesterol, high-sensitivity C-reactive protein, insulin and waist circumference [172]. These findings are in line with the above-mentioned cross-sectional study results for 187 Polish vegan children, in which a vegan diet was associated with a healthier CVD risk profile but with relative differences in growth (e.g., 3 cm shorter and 4–6% lower bone mineral content) and specific nutritional status compared to an omnivorous diet [166]. Notably, concerns about the inadequacy of some nutrients in a vegan diet, especially in early life, are of great importance. However, society faces greater challenges related to the consequences of an obesogenic food environment that fuels current major public health epidemics (e.g., obesity, type 2 diabetes and common chronic disease) with a food-addictive dietary pattern consisting of an excess of energy, highly processed carbohydrates, free sugar, saturated and total fat and cholesterol [173,174].

During the metabolic programming period, maternal nutritional behavior represents the crucial component for favourable health outcomes. We need more high-quality evidence involving the prenatal, pregnancy, infancy and childhood periods (for which the main challenge is ethics and adherence). To date, however, we have sufficient overall evidence that a well-designed vegan diet (e.g., with an adequate intake of nutrients) that is properly medically supervised should be considered safe and healthy for pregnancy, infancy and childhood [23,25,153,160,175,176]. In contrast, an unbalanced vegan diet that lacks energy and micronutrients should not be seen as protective in this sensitive window for nutritional programming and metabolic imprinting in terms of later beneficial outcomes [156,177].

## 4. Vegan Diet and Environmental Footprint

The EAT-Lancet Commission convened 37 leading scientists from 16 countries in various disciplines and stated that food is the single strongest lever to optimize human health and environmental sustainability on Earth. Thus, the recommendation of the commission was to increase the consumption of plant-based foods (i.e., fruits, vegetables, nuts, seeds and whole grains) while substantially limiting animal-based foods [1]. Similarly, 11,258 scientists from 153 countries encouraged the eating of mostly plant-based foods while reducing the global consumption of animal-based products to improve individual health and to significantly lower emissions of greenhouse gases (GHGs), including methane [178].

At least one-third of global anthropogenic GHG emissions are attributed to the current food system (e.g., as it exists now), whereas the largest contribution comes from agriculture and land use/land-change activities (71%) [179]. In a recent systematic review that examined the changes in GHG emissions as well as land and water use as a result of shifting current dietary intakes to environmentally more sustainable dietary patterns, it was shown that a vegan diet has the greatest impact on the reduction in GHG emissions [180]. In addition, German study examined the external climate cost using life-cycle assessment and meta-analytical approaches and showed that organic plant-based products were associated with the lowest external GHG emission cost [181].

In conclusion, a systematic review of 16 studies and 18 reviews that address the important question of which diet has the least environmental impact suggests that for the environment, a vegan diet may be the optimal diet [182].

## 5. Nutrients of Concern in Vegan Diet

There are several nutrients of particular concern in a vegan diet. Importantly, we are aware that few of these concerns are legitimate, while others are disputable when we talk about a well-designed vegan diet in which the majority of calories come from whole-food, plant-based foods.

A recent systematic review on the dietary intake of European vegans revealed several nutrient deficiency concerns [183]. The review included 48 studies (12 cohorts and 36 cross-sectional), of which 6 cohorts and 16 cross-sectional studies were published 15 or more years ago. Although every review may be of great importance, in this case, several nutrients of concern may be irrelevant today, as we have increasingly accessible research results on the well-designed vegan diet and books, lectures, podcasts and free guides that are more accessible to the general population.

The recommended values for dietary intake in Slovenia are based on Central European D-A-CH (German [D], Austrian [A], and Swiss [CH] reference) values, while recommendations for the intake of EPA/DHA are taken from the European Food and Safety Authority (EFSA) [184,185,186]. Concerns about the nutritional adequacy of vegan diet intake may be divided into three groups:sufficient intake of energy and protein;nutrients that should be obtained from dietary supplements or enriched foods (i.e., vitamin B12, vitamin D, omega-3 fatty acids); andmicronutrients of concern (i.e., calcium, iron, zinc, iodine and selenium).

### 5.1. Energy and Protein Intake Concerns

People on a vegan diet must take into consideration that an unprocessed or minimally processed vegan diet may be low(er) in energy if it is not well-designed. In several interventional studies, successfully using a vegan diet for CVD risk factors or diabetes type 2 management for obese and overweight patients might involve potential concerns about a lower average energy intake (1315, 1422 and 1450 kcal/day) [60,114,118], which can result in certain micronutrient inadequacies (additional to the nutrients that must be obtained via supplementation). Furthermore, in two of the three recently published studies mentioned, the authors did not report micronutrient intake values (i.e., they studied other outcomes) [60,118], whereas in earlier long-term interventional studies (74 weeks), the authors published detailed dietary intake reports, which confirmed nutrient inadequacy for calcium intake but not for iron and zinc intake [114] based on valid dietary guidelines for healthy people on an omnivorous diet. Other researchers have analysed the dietary intake of a well-designed, very low-fat, supplemented vegan diet (e.g., with soy protein, selected vitamins and minerals, nutrient-fortified plant-based foods and EPA/DHA from fish oil supplementation) that was successfully used in the “treatment” of early-stage prostate cancer. For this group of patients (PA was included), the researchers reported higher energy (2125 kcal/day) and protein intake (108 g/day or 20% of energy) and the consequent nutritional adequacy of micronutrients of concern (i.e., calcium, iron and zinc) [187]. Furthermore, in the last decade, in numerous cross-sectional studies on participants who ate vegan diets, authors reported higher energy intake (i.e., from 1791 to 2832 kcal/day) [98,143,188,189,190,191,192,193,194,195], which is in line with recommendations for including moderate-intensity PA levels as part of healthy and active lifestyles [184,196]. Notably, higher energy intake per se does not automatically result in nutritional adequacy of nutrients of concern (i.e., a higher energy intake might be easier to achieve by eating an ultra-processed vegan diet that is deficient in fiber and micronutrient content, for example, with excessive usage of vegetable oils, refined grains, fruit juice or margarine, which are generally very nutrient-depleted foods). In conclusion, based on the results of nutritional adequacy in intervention studies and (comparative) cross-sectional studies, we take the position that a well-designed vegan diet is easily energy adequate. However, proper meal composition and intake to satiety (*ad libitum*) is extremely important and may be different from Western-type eating in obesogenic food environments [73,174,197].

A recent review reported that average protein and amino acid intake from vegan diets ranged from 62 g/day to 82 g/day [198], which is in line with reference values for protein intake for the general adult population and for elderly individuals set to 0.8 g and 1.0 g per kg body mass/day (48–57 g/day and 57–67 g/day) [184,196]. In brief, vegan diets typically meet or even exceed reference values for protein intake, especially when energy intake is adequate and based on whole-food sources [98,143,187,188,190,191,193,194,195,199]. These data are not surprising, since plant-based protein sources are readily available in protein-rich food groups, such as legumes, (intact) whole grains, nuts and seeds (butters) as whole-food sources, and in plant-protein supplements and plant-based meal replacements. Regardless, there is a prejudice that plant-based protein sources are not of the same quality as animal sources. Several new review studies explain that restricting the intake certain amino acids (i.e., methionine, tryptophan and leucine) and overall protein intake, which was once considered a limitation of a vegan diet, is recognized as beneficial, as it generally decreases ageing-related comorbidities and thereby increases health and life-span [200,201]. Furthermore, there is also a prejudice that plant-based protein sources do not enable maximum muscle synthesis as animal-based proteins do and that this can only be achieved through protein supplementation; however, these arguments are not supported by the facts. The required sufficient total protein intake (approximately 1.6 g per kg body mass/day) and protein intake per meal for maximum muscle synthesis is between 20–25 g, thus providing at least 1.2–3 g of leucine [202], which is fairly easy to achieve if there is a need; for example, a meal with 60 g of lentils and 60 g of buckwheat porridge exceeds 20 g of protein, while a meal with 100 g of whole grain spaghetti, 50 g of soy flakes/70 g of soy tofu and 100 g of corn contains approximately 25 g of protein [203]). In addition, sufficient intake of the amino acid leucin has been proposed as a key factor to trigger the muscle growth response. However, most plant-based protein sources are similarly rich in leucine (i.e., average ranging from 6–12 g/100 g) compared to animal protein sources (i.e., average ranging from 8–12 g/100 g) [198]. This often-advocated belief is based on a system of determining the “Digestible Indispensable Amino Acid Score” (DIAAS), where animal-based proteins (e.g., meat and milk) exhibited greater DIAAS values in the reference for adults than plant-based proteins, with the exception of potatoes and soy protein [142,204].

We emphasize that this remains a controversial method of assessing the quality of plant- and animal-protein sources. Even with the DIAAS method, this process maintains several limitations that are specifically related to plant-based dietary patterns. These include failure to translate nitrogen differences into protein-conversion factors between plant- and animal-based foods; limited representation of commonly consumed plant-based foods in larger quantities (e.g., fruits, vegetables, certain legumes, nuts and seeds), thus contributing to underestimation of the actual digestibility of plant sources of protein in humans eating strict plant diets within the DIAAS framework; inadequate recognition of increased digestibility associated with frequently consumed heat-treated and processed foods; formulation centered on fast-growing animal models rather than humans; and a focus on individual isolated ingredients compared to the vegan nutritional matrix. All of these factors reduce its application both at rest and in an exercise setting [205].

Regardless, the protein quality of a single ingredient should be of less importance than the quality of the well-designed mixed (vegan) meal that is normally consumed [20,204,205]. Various strategies may be applied to increase the anabolic properties of plant-based protein if there is a justified need, including (i) consuming higher amounts of and/or multiple plant-based protein-rich sources at a meal or (ii) using plant-based protein supplements [19,20]. With regard to the latter, the results of well-designed studies demonstrated limited eligibility for the consumption of protein supplements for muscle growth or strength when performing resistance training if protein intake from a conventional diet has already met the recommendations [206,207]. Notably, not all adults, physically active people or even most competitive athletes want or need to achieve maximum muscle synthesis for sport performance. Muscle mass and strength are primarily built by resistance training [208]. Therefore, to achieve the recommended protein intake, a vegan diet needs to be well-designed, which includes the appropriate composition of a meal plan and a sufficient amount eaten. Last, it must be emphasized that on a well-designed vegan diet, there is no need to “consciously” combine different plant-protein sources in each meal on a given day [198,209]. However, even if it happens that a particular vegan meal is not optimally composed, the individual has a supply of free amino acid “pools” in body tissues (e.g., mostly in skeletal muscle and body fluids) in the amount of approximately 70–90 g [210,211].

Some amino acids, such as taurine, carnosine, creatine and anserine, are almost exclusively found in foods of animal origin [212]. Thus, some authors conclude that since these nutrients are absent from all plants (e.g., grains, legumes, potatoes and nuts) [213], vegans are at great risk of deficiencies, particularly if they are physically active or if they regularly perform intensive exercise [148]. Currently, however, health and sport-performance efficacy and safety have been well established only for creatine [214,215]. In addition, vegan athletes may benefit even more than omnivores (e.g., improved exercise performance); however, at present, the research is inconclusive on whether an increased creatine capacity translates to a greater increase in performance in vegan athletes than among their omnivore peers [149]. Furthermore, β-alanine supplementation has been shown to increase muscle carnosine concentrations, leading to improvements in high-intensity exercise performance [148,212]. However, research is clearly lacking on, for example, β-alanine and taurine in physically active or competitive vegan athletes [148,216]. Moreover, taurine is an essential amino acid for preterm neonates that is provided by breast milk [217] and is a potential ergogenic aid for preventing muscle damage, attenuating muscle protein catabolism, decreasing oxidative stress and improving performance in endurance athletes [148,212]. Further research is needed to validate the health and performance role of these other amino acids (e.g., need, efficacy and safety) for the general adult vegan population and in physically active or competitive vegan athletes.

### 5.2. Nutrients That Should Be Obtained from Dietary Supplements or Enriched Foods

#### 5.2.1. Vitamin B_12_

Vitamin B_12_ deficiency is a common global nutritional problem in people of all ages, especially among elderly individuals [218,219,220]. Furthermore, adequate intake of vitamin B_12_ is seen as the greatest concern in vegan diets. As vitamin B_12_ is not synthesized by plants or by (vegan) animals, it needs to be supplemented regularly with reliable sources [23]. For adults (16–65 years), the reference value is set to 4 µg/day [184,186]. However, because of substantial differences in its absorption rate, which depends on intake, age, BMI, health status and oral dosage (e.g., preferable in the form of cyanocobalamin [221], except in impaired renal function [222]), 100–200 µg of B_12_/day or 2000–4000 µg B_12_/*w*, divided into 2–4 doses/week, is recommended. A high intake recommendation compared to the reference value takes into account the efficiency of absorption and passive absorption routes [222,223,224]. Finally, routine laboratory testing (i.e., at least serum vitamin B_12_ status and, if possible, methylmalonic acid (MMA) and total plasma homocysteine status, which are considered better markers of vitamin B_12_ deficiency than serum vitamin B_12_), is warranted to detect a possible vitamin B_12_ deficiency [223].

#### 5.2.2. Vitamin D

Vitamin D deficiency is another major nutritional public health issue worldwide and is common among people of all ages. This might be the case even in countries with low latitudes, where it has been generally assumed that because of an abundance of UVB radiation, vitamin D synthesis is sufficient [225]. However, it seems that vitamin D deficiency is primarily a consequence of today’s institutional lifestyle. Furthermore, due to the high incidence of vitamin D deficiency in the autumn-winter period (80% of adult and healthy Slovenes have vitamin D (25-hydroxycytamin D) insufficiency, and 40% have severe vitamin D deficiency, with an average vitamin D intake of only 3 µg/day (for adults, the reference value is set to 20 µg/day) [226,227]).

It is generally accepted that the primary vitamin D sources are sunlight (with a sufficient UVB index), diet and dietary supplements [228]. It is practically impossible to meet recommendations for vitamin D (20 µg/day) exclusively by preformed foods (i.e., fish liver oil, oily fish, liver and, in smaller doses, meat and egg yolk). According to Slovenian data (e.g., Open Platform for Clinical Nutrition, the online application tool with an extensive food/dish database), to provide enough vitamin D exclusively through diet, an individual should regularly consume large amounts of wild salmon (i.e., 17 µg vitamin D/150 g), egg yolks (i.e., 1 µg vitamin D/100 g), whole-fat milk (<1 µg vitamin D/7 dcl) and chicken liver (1 µg vitamin D/100 g) [203]. Furthermore, vitamin D-rich foods normally contain high amounts of saturated fat, cholesterol and possibly various environmental contaminants [41,229].

The status of vitamin D was found to be lower among vegans than omnivores when serum vitamin D (i.e., 25-hydroxyvitamin D) levels were examined among vegans living at high geographical latitudes or in winter or early spring months [23,230]. However, based on data from cross-sectional studies comparing the dietary intake of vitamin D intake among vegans or omnivores [189,226,231] and the current serum vitamin D status of adults and healthy Slovenes [227] or high-performance-level Slovenian athletes [232], we may conclude that the problem of vitamin D deficiency does not originate in dietary patterns. Furthermore, researchers in Adventist Health Study 2 examined vitamin D intake relative to serum vitamin D status and could not confirm a significant difference in serum vitamin D levels between vegetarians (e.g., those who ate meat and/or fish < 1 time/month) and nonvegetarians [233].

To conclude, in addition to safe, sufficient and regular exposure to sunlight in the spring and summer months (with a sufficiently high UBV index), adequate and regular vitamin D supplementation regardless of diet pattern is recommended in the autumn and winter months (October-April) with 800–2000 IU/day (for pregnant women with 1500–2000 IU/day) [234]. Moreover, for people with suboptimal vitamin D status or older or obese individuals, vitamin D intake should be increased under the supervision of a personal physician [235,236].

#### 5.2.3. Omega-3 Fatty Acids

To ensure sufficient intake of omega-3 fatty acids, a well-designed vegan diet should include proper amounts of ALA and EPA/DHA. ALA is synthesized by plants, while EPA/DHA is synthesized by phytoplankton. Plant-based foods are particularly good sources of ALA (e.g., flaxseeds, chia and hemp seeds, rapeseed oil, walnuts and soy). Furthermore, ALA may endogenously be converted to EPA/DHA; however, the conversion rate is slow and inefficient and affected by FADS1 and FADS2 polymorphisms, intake of omega-6 fatty acids, lifestyle and health status characteristics (i.e., diet, age, sex, smoking, alcohol and chronic diseases) [237,238,239,240,241].

Due to great genetic variability, differences in lifestyle and health status, advancing age and pregnancy and lactation requirements, supplementation with EPA/DHA may be warranted. Direct intake of EPA/DHA can be ensured by eating fish or supplements (i.e., from fish oils or other marine sources and cultured from microalgae) [240,242]. Compared to omnivores, vegans typically have lower serum and plasma levels of EPA/DHA [190,237,243], which may be due to different intakes of ALA, EPA/DHA and to the numerous factors discussed above. However, the EPIC-Norfolk study results showed that vegans had comparable plasma levels of EPA/DHA to omnivores, regardless of the fact that they did not additionally take in EPA/DHA [239].

It is suggested to consume at least one tablespoon per day of flaxseed or chia seeds to obtain an adequate intake of ALA [237,244]. In addition, given the inconclusive evidence, for vegans with reduced conversion ability, for vegans who have greater need and to maintain long-term cognitive health, 250 mg/day pollution-free EPA/DHA (i.e., microalgae, fish oil and other marine sources, yeast) should be supplemented in addition to an adequate intake of ALA [185,237,245,246]. Furthermore, for pregnant and lactating mothers who do not eat fish or other marine sources, experts recommend 300 mg/d DHA [242].

### 5.3. Micronutrients of Concern (i.e., Calcium, Iron, Zinc, Iodine and Selenium)

#### 5.3.1. Calcium

For adults (of both sexes) in Slovenia, an adequate intake of calcium is set to 1000 mg/day [184,186]. According to the results of recent European cross-sectional studies, this quantity can be easily met with a well-designed vegan diet or is comparable to intake on an omnivorous diet [71,98,190,191,199]; however, other studies disagree [188,189,192,193,195,231,247]. Notably, omnivore groups in these studies also did not reach the calcium intake recommendation [192,194,231,248].

In many respects, this variability suggests that adequate calcium intake is a general problem of proper meal planning rather than a problem of a vegan diet per se. Although still speculative, the evidence of calcium intake among vegan diets is in line with the results of global dietary calcium intake among general adults [249]. Due to known paradoxical results of observational studies where high calcium intake was found to be associated with a high prevalence of hip fracture, and due to the serious limitations of most interventional studies (i.e., short duration, high habitual calcium intake, the method used), the reference for adults to maintain appropriate bone health is set differently by the World Health Organization (500 mg/day), the United Kingdom (700 mg/day) and the United States (1000 mg/day) [250].

Interestingly, the results of a recent German cross-sectional study that compared the bone health of adult vegans (*n* = 36) and omnivores (*n* = 36) found a 5% difference in one measurement of bone health status. However, this small study should not be considered clinically relevant to the drawing of solid conclusions since the results of quantitative ultrasound measures were not adjusted for BMI and the sub-analysis showed a trend of bone health improvement with the duration of a vegan diet [251]. Our speculative interpretation is in line with the results of the current review, which also suggests that there is no evidence that a proper vegan diet, when carefully designed to maintain adequate calcium and vitamin D intake, has detrimental effects on bone health and, in fact, may have beneficial effects [252]. The importance of a well-designed vegan diet was further suggested by findings in a recent Epic Oxford study on UK vegans. In this study, it was found that the fracture rates among vegans were higher than those among omnivores even after controlling for BMI, calcium intake and bone mineral density [253]. This concern was probably due to a combination of low calcium intake (611 mg/day and 580 mg/day for men and women), low vitamin D status (e.g., high geographical latitude “condition” supplementation [230]) and, perhaps, vitamin B_12_ inadequacy and very low PA levels. In addition, the UK vegan diet was poorly designed because fiber intake was below 30 g/day, whereas fat intake was close to 30% of total energy intake; therefore, these results should not be a reference source for well-designed vegan diets [253].

Unfortunately, serious conclusions on the basis of a recent systematic review with a meta-analysis of 17 cross-sectional studies, where researchers examined the association between vegetarian diets (including a vegan diet) and an omnivorous diet on bone health, are not possible. The reasons for this are that only five studies on a vegan diet were included in the review, of which one study included raw vegans, another included Buddhist nuns and one older adult (aged 70–89 years), where the protein intake of the subjects on a vegan diet was only 35 g/day, while two performed comparable bone mineral density to the subjects on the omnivore diet [254]. Finally, with regard to the association between calcium and bone health, resistance training might have much more importance [255].

Furthermore, the problem of calcium absorption from plant food sources is often mentioned due to the higher content of oxalates (and phytic acid in legumes, cereals and nuts/seeds, for example) in some sources (e.g., spinach) but not in other sources (e.g., broccoli, kale or kiwi), where calcium absorption from broccoli is comparable to milk (41% from broccoli vs. 32% from milk) [256,257,258]. Regardless, according to the above-mentioned results of recent European cross-sectional studies that did not estimate an adequate calcium intake among adult vegans [188,189,192,193,195,231,247], it seems to be often a great challenge to consume enough calcium-rich plant foods every day (in comparison, drinking two to three cups of cow’s milk seems much easier). Importantly, in addition to a sufficient total intake of calcium from any plant food source, we also paid attention to the regular intake of a sufficient amount of dark leafy greens and/or cruciferous vegetables and fruits.

In conclusion, a well-designed vegan diet that includes sufficient and regular consumption of plant-based foods with a high calcium content (e.g., cruciferous and green leafy vegetables with spinach as an outlier for the high oxalate content, legumes and seeds, especially tempeh and calcium-set tofu, as an outlier for the phytate content, and dried figs), regardless of the absorption rate, while avoiding excess salt intake, may provide adequate calcium intake.

#### 5.3.2. Iron and Zinc

Several of the frequently mentioned micronutrient intakes of concern (i.e., iron and zinc inadequacy) in a vegan diet have also been highlighted in past review studies. In contrast, cross-sectional research published in the US and European countries in the last 15 years did not confirm the aforementioned insufficient intake of iron (i.e., adequate intake for females and males is 10–15 mg/day and 10 mg/day [196]) and zinc (i.e., adequate intake for females and males is 7–10 mg/day and 11–16 mg/day, depending on phytate intake [196]) [143,188,189,191,193,195,231,247].

It is widely known that plant-based iron (i.e., nonheme iron) is not absorbed as efficiently as heme iron from animal-based foods. However, avoidance or restriction of the intake of heme iron is considered beneficial (e.g., more carefully controlled to avoid overload) [259,260,261,262]. Nevertheless, nonheme iron absorption from plant-based foods can be significantly enhanced (even over 300%) by adding vitamin C-rich foods to the meal (e.g., fruits, bell pepper, broccoli, vinegar) [263]. In addition, adding garlic and onion to grains or legumes may significantly increase the bioaccessibility of iron by up to 66% in grains and up to 73% in legumes in both their raw and cooked forms [264]. It is also important that the consumption of herbal teas (i.e., black and green), coffee, cocoa or red wines is at least one hour apart from a meal [262,265].

In a number of cross-sectional studies over the last 10 years, researchers also found that zinc intake among vegans was adequate; in addition, vegans performed much better than omnivores in regard to zinc intake [143,188,189,191,193,195,231]. Nevertheless, a well-designed, high-fiber vegan diet (e.g., a diet with high intake of whole grains, legumes, fruits and vegetables) may require a higher zinc intake due to high phytate content and/or a high calcium intake that partially inhibits its absorption [196,266]. Similar to iron, adding both garlic and onion to grains or legumes may significantly increase the bioaccessibility of zinc by up to 159% in grains and up to 50% in legumes in both their raw and cooked forms [264]. In addition, zinc-containing plant foods (e.g., legumes, grains, nuts and seeds) can be further increased by appropriate methods of preparation (e.g., soaking, germination or fermentation) of certain plant food groups [267].

#### 5.3.3. Iodine and Selenium

Worldwide, iodine deficiency disorder is considered to be a major nutritional challenge that manifests as a health problem that affects all groups of people [268]. Nevertheless, vegans appear to have an increased risk of low iodine status [247,269]; therefore, a well-designed vegan diet should properly address this challenge.

The reference value for adequate iodine intake for adults is 150 µg/day [184,186], which cannot be easily achieved without exceeding the permitted reference for daily salt intake [270,271]. According to the most recent estimates on iodine status, published in 2012, the proportion of the Slovenian population with an iodine level lower than 100 µg/L was 23% [272,273]. These data are of great importance, since excessive sodium intake (e.g., 40% of salt is sodium) is a leading dietary risk factor associated with elevated blood pressure, CVD and premature death [274]. Moreover, up to 80% of dietary salt is usually consumed with processed foods [275], especially baked goods, cheeses and processed meats [276]. Because a well-designed vegan diet does not contain these food groups, it may include added iodized salt within food preparation (e.g., in salads, vegan burgers, pastas, potatoes, risotto, soups and sauce), used with caution if various high-salt additives are included in food preparation (e.g., soy or tamari sauce) or pre-prepared meals (e.g., salted canned legumes or salted nuts). Importantly, we must take into account that during boiling (37%), pressure cooking (22%), steaming (20%) and roasting (6%), iodine losses occur [277], so it is reasonable to adjust cooking methods to minimize iodine losses (i.e., seasoning with salt at the end of preparation). In addition, seaweed (e.g., nori, awake, kelp), a naturally representative-rich source of iodine, is not consumed very often in Slovenia. Nevertheless, in the case of a regular intake of seaweed (e.g., particularly kombu), excessive intake of iodine should be considered (there is an urgent need to improve the database on the nutrient content of iodine in various foods and products). In addition, plant-based food sources usually contain small amounts of iodine (e.g., potatoes with peels, berries and legumes), so their consumption in greater amounts is encouraged. However, current recommendations for salt fortification are suggested to be revised [270], or appropriate iodine supplementation might be suggested.

For adults (of both sexes) in Slovenia, adequate intake of selenium for females and males is set to 60 µg/day and 70 µg/day [196]. According to the results of recent European cross-sectional studies, this quantity can be easily met with a well-designed vegan diet or is comparable to selenium intake on an omnivore diet [71,190,231,247]; however, other studies disagree [195,248]. Importantly, several cross-sectional studies evaluating the dietary intake of vegans did not report selenium intake [188,189,191,192,199], which suggests that there is an urgent need to improve the database on the nutrient content of selenium in various foods and products. Regardless, plant-based sources rich in selenium are whole grains, legumes, vegetables, nuts and seeds [39,278]. In addition, one Brazil nut a day, a naturally representative rich source of selenium, is an ideal way of meeting selenium recommendations [39,278].

## 6. Practical Recommendations for Implementing Vegan Diet

A well-designed vegan diet is suitable in general for normally active adults. It should provide sufficient energy and nutrient intake, which can be achieved with appropriate weekly and daily meal planning using primarily nutrient-dense, unprocessed or minimally processed plant-based foods.

Based on official Slovenian and different European recommendations for generally healthy adults [184,185,186,279,280] and the (review) studies on a vegan diet [23,24,56,58,60,67,71,73,100,104,114,124,137,140,148,199,281], we suggest the main points of a well-designed vegan diet for healthy adults. It should be characterized by a high intake of fiber (≥45–60 g/day), a moderate intake of protein (<15% of energy), and a low intake of total fat (10–20% of energy), saturated fat (≤5% of energy), free sugars (<5% of energy) and sodium (1500 mg/day). The required supplementation must include reliable sources of vitamin B_12_ for the whole year and vitamin D in the winter months (vitamin fortified foods and/or a supplement), and optionally may include EPA/DHA throughout the year. Therefore, we recommend a well-designed vegan diet for generally healthy adults:unprocessed or minimally processed plant-based foods, especially whole grains, legumes, fruits and vegetables, can be eaten *ad libitum* (i.e., moderate to full satiety) [197];whole grains (e.g., oats, buckwheat, rice, millet, quinoa, corn, rye), legumes (e.g., beans, lentils, soy foods, peas, chickpeas) and fruits (e.g., all types but preferably berries) should provide the majority of energy intake and should be included in most meals [41];tubers (e.g., white and sweet potatoes), colorful, green leafy and cruciferous vegetables, fresh/dry herbs, spices and aromatics are encouraged on a daily basis [41];nuts (e.g., walnuts, hazelnuts, almonds, peanuts) and seeds (e.g., flaxseeds, chia, sesame and hemp seeds) should be consumed daily but sparingly within a meal and in a natural form (without added salt or oil and, if possible, unroasted) [282];plant-based or non-dairy milk alternatives (with vitamin D, B_12_ and calcium when possible): (a) cereal based: oat milk, rice milk, corn milk, spelt milk; (b) legume based: soy milk, peanut milk, lupin milk, cowpea milk; (c) nut based: almond milk, coconut milk, hazelnut milk, pistachio milk, walnut milk; (d) seed based: sesame milk, flax milk, hemp milk, sunflower milk; and (e) pseudo-cereal based: quinoa milk, teff milk and amaranth milk are optional and, whenever possible, should have no free sugars, added vegetable oil or salt [283,284];to enhance the aroma and flavor as well as the antioxidative potential of dishes, culturally inspired spice combinations are recommended: fresh and/or dried herbs, spices and aromatics, such as Mediterranean, which is comprised of basil, garlic, leek, marjoram, onion, oregano, rosemary, sage, thyme and white pepper. Other options are Thai (i.e., chili, ginger), Mexican and Spanish (i.e., chili, coriander, cumin, parsley), Moroccan and African (i.e., cardamom, clove, saffron), Japanese and Chinese (i.e., wasabi), Turkish and Greek (i.e., anise, cilantro, chili, thyme) and Indian (i.e., curry, turmeric) [285];vegetable oils/fats (e.g., sunflower oil, coconut and palm fat) are not recommended [286,287,288]; olive oil and canola oil may be used sparingly in salad dressings to bridge the transition phase or for higher energy requirements;for seasoning, lemon and a variety of vinegars are recommended, along with the use of iodized sea salt [23,197];primary liquids should be water and herbal teas (e.g., green, black and hibiscus), and to a lesser extent fresh green smoothies, whereas other liquids (e.g., sports drinks, plant-based protein supplements in connection with PA, plant-based meal replacements) may be used in cases of increased energy and nutritional requirements (e.g., with the precondition of maintaining an intake of 45–80 g fiber/day, which is considered for a well-designed vegan diet [71,187,191,199,289,290], the recommended daily intake of protein and micronutrients [196]) and with as low as possible an intake of free sugars [280] and saturated fats [291];to meet higher energy requirements, individuals may incorporate carbohydrate-rich foods (e.g., dry fruits, whole grain spaghetti, polenta, whole grain bread) in smaller amounts of fat-rich foods (e.g., nuts, seeds and avocado) or mixed sources (e.g., burgers, spreads and dressings, ideally without free sugars and fat);the main appropriate cooking methods are moist heat (such as poaching, simmering, boiling, steaming), combination cooking (braising, stewing, pressure cooking), no heat (curing, culturing, fermenting, acidifying, sprouting, soaking, high-speed blending, pureeing, vacuum sealing, juicing (in rare cases, small amounts only), dehydrating) and dry heat (air drying/dehydrating, sweating, searing, stir-frying, griddle cooking, baking, roasting, grilling, broiling, sauntering) [292].

Adherence is key in pursuing the effects of a long-term diet given the importance of a day-to-day basis. There is a notion that adopting a vegan diet can be quite demanding, especially since many individuals may face significant problems, social pressure and obstacles in achieving this goal [68]. However, dietary adherence/acceptability was tested in a relatively small, six-month randomized controlled trial (“the New Diets study”), which showed that adherence/acceptability across five different diets, including a vegan diet, was very similar [293]. Self-efficacy, social identification, improved health, body mass/body composition and participation in caring and supportive programs that provide monitoring and relevant feedback progress may further improve long-term adherence [57,62,98,100,101,114,124,294,295,296].

By adopting a well-designed vegan diet (e.g., high nutrient density), the uncomfortable physical and emotional symptoms of hunger become much less prevalent, and adherence to a healthy diet regime is increased [297]. Informed choice, emotional support, increased accessibility of healthy and affordable food systems and healthy and tasty food preparation methods, especially when eating out, are important determinants that give people the opportunity to improve the quality of their lives [297,298].

In addition, the vegan diet is compatible with sports performance. It has the potential to encourage active people and health and sport experts to be more tolerant when a person expresses interest in implementing a vegan diet. When following a vegan diet for regular exercise or sports performance, the diet needs to be planned carefully, especially in terms of providing (i) enough energy and (ii) proportionally more nutrients. In addition, a well-designed vegan diet for physically active individuals and athletes should primarily be based on conventional unprocessed or minimally processed plant-based foods to ensure the majority of energy and nutrient needs are met. However, the available data suggest that further supplementation in addition to vitamin B_12_, D, and EPA/DHA (e.g., plant-protein supplement, plant-based meal replacement, sport drinks and creatine) may offer health and performance benefits or may facilitate individuals’ adherence to a vegan diet [16,23,71,137,140,143,145,148,149].

## 7. Barriers to the Implementation of the Vegan Diet/Lifestyle

The development of a vegan diet as a respected nutritional paradigm faces many barriers that can be divided into at least six groups: (1) lack of education in the vegan field of nutrition by physicians and dietitians, as well as in connection with professional (e.g., conflict of interest) and personal (e.g., eating according to another dietary pattern) “contamination”, (2) lack of interest in researching the vegan diet, (3) an obesogenic food environment that increases challenges, (4) believing that a vegan diet is expensive, (5) specific characteristics of a vegan diet/lifestyle and (6) animal-based analogues/alternatives.

### 7.1. Lack of Education in the Vegan Field of Nutrition by Physicians and Dietitians

In Europe [299,300,301,302], in the United States’ medical schools [303,304,305] and elsewhere [306,307,308], an important lack of education about (clinical) nutrition in general is observed. Ironically, we often hear that “nutrition has little or even no impact on health and chronic diseases” [309]. This social situation is rhetorically extremely interesting since poor dietary habits worldwide are the leading contributor to a range of chronic noncommunicable diseases [87,274]. Given this situation, ignoring nutrition or not including advice about healthful eating as a routine part of 21st century medical practice should no longer be defensible [308,310].

However, the gradual wider adoption of a vegan diet in official medicine is also reflected in the news from Wayne State University School of Medicine’s Detroit School of Medicine, which prescribes a mandatory four-week whole-food plant-based nutrition education for medical students that also includes interactive cooking presentations [311]. In addition to this encouraging information, the University of Florida’s Shands Hospital is one of the first healthcare institutions in the United States that provide vegan meal options in inpatient settings (five different options for breakfast, three for lunch, three for dinner and additional chef specialties (e.g., patients also receive educational materials about the role of diet and lifestyle choices and their impact on chronic illnesses) [312], followed by a number of hospitals in the United States that have started inpatient 100% vegan menus [313].

Additionally, limited data have shown that on average, there is a wide discrepancy between an (unhealthy) lifestyle and the (unhealthy) appearance of the medical profession on the one hand, and what the medical profession (i.e., the authority in the field of health) is supposed to represent in society on the other [314]. Importantly, a lack of knowledge about nutrition often translates into unhealthy dietary patterns and often manifests as inadequate eating habits in professionals themselves [314,315]. Too often, educational resources about nutrition contain one-sided information about a vegan diet, often involving negative and ridiculous connotations and prejudices (e.g., generalizations that it is unnatural and nutritionally inadequate (e.g., because it is necessary to supplement it with vitamin B_12_), difficulty maintaining this behavior, inconsistent with modern dietary guidelines and that a vegan diet is eaten mainly by people with ethical or environmental motives). However, students should be taught about both the health benefits of a well-designed vegan diet and the potential risks of a poorly designed vegan diet in a respectful and productive manner [316].

In conclusion, we suggest that both clinical nutrition and a mandatory vegan diet should be an essential part of common undergraduate and postgraduate education in all faculties teaching human nutrition (i.e., medical faculty, biotechnical faculty, faculty of sport, faculty of health sciences).

### 7.2. Lack of Financial Interest in Research on the Vegan Diet

On 22 November 2021, the www.clinicaltrials.gov register [317] included 50 studies available with the “vegan diet” search filter; however, only three studies (6%) were industry funded. For comparison, for CVD (e.g., “cardiovascular drug” search filter only) or type 2 diabetes (e.g., “diabetes type 2 drug” search filter only), both of which a well-designed vegan diet may prevent or reverse [67,99,100,101,102,114,115,116,117,118], we found 8506 (4382 (51%) industry funded) studies, respectively.

In addition, public popularity, as depicted by a Google Trends search for the terms “vegan”, “vegetarian” and “meat” in Germany, the US and the UK for the period from 2004 to 2019, showed that people are looking for more information on topics related to the term “vegan” [318]. Importantly, with regard to plant-based dietary patterns, the frequency of publication on PubMed from 2000 to 2019 showed that vegan-, vegetarian- and plant-based search terms increased significantly over the years, but more interest was seen for vegetarian-related terms [318]. This may be due to the greater interest of the industry in the popular lacto-vegetarian and Mediterranean diets, which are included within the plant-based diet group. Regardless, both views underscore the increased interest in vegan or plant-based diets in general.

As long as a healthy vegan diet is not part of the healthcare system reward financial framework, nutrition as a science (especially science on vegan diets) cannot receive the respect it deserves or will remain neglected in the medical community with its current status, which slows the development of a vegan lifestyle [315]. In addition, the presented data pose a serious threat not only to the more “extensive” development of the vegan diet within the field of nutrition but also in terms of the objective dissemination of information and the accessibility of healthy vegan meals outside the home environment. Those responsible for nutrition education are encouraged to include a vegan diet in the curriculum, preferably from an expert who will be able to objectively represent the content of this important profession.

### 7.3. Obesogenic Food Environment That Poses an Increasing Challenge

Industry practices that foster an obesogenic food environment with foods that can be labelled unhealthy, energy-dense, ultra-processed, additives rich and micronutrient deficient are likely the key external contributors to addictive-like overeating and obesity [174,297,298].

In addition, there is a lack of healthy, low-fat, vegan meals on menus in restaurants (e.g., not just a salad bar), canteens and schools, and these meals might be very welcome for anyone on an omnivorous diet or a vegetarian diet, as well as those with allergies (e.g., to milk, eggs, fish or lactose).

Furthermore, the state could subsidize these meals and thus encourage people to consume more plant-based foods, which is very beneficial for health and the environment (i.e., within the framework of climate goals), and is ethical towards animals.

### 7.4. Believing That a Vegan Diet Is Expensive

There is a belief that a vegan diet is expensive, especially if we consider a well-designed vegan diet (e.g., foods that are less dense in energy usually require a higher food volume intake), consumption of mostly organically certified foods or products and eating outside the home often. However, the researchers in one study outside a clinical trial (the GEICO study) setting found that the vegan group reported a decrease in food cost compared with the control group, while reporting a 40–46% decrease in health-related productivity impairments at work and in regular daily activities [319]. The current state (e.g., December 2021) of food prices in Slovenia in the most commonly available grocery stores also indicates that a well-designed vegan diet can be affordable compared to expenditures on commonly consumed food. In support of this, the prices of oatmeal (1.1–1.5 €/kg), buckwheat (2.4–3 €/kg), wholegrain rice (3.5 €/kg), wholegrain pasta (1.2–3.4 €/kg), legumes (beans (2–3 €/kg), lentils (1.8–2.8 €/kg), soy tofu (4.9–7 €/kg), nuts and seeds (flaxseeds (2.8 €/kg), unshelled sesame seeds (4.8 €/kg)), dates (7.2 €/kg), potatoes (0.8 €/kg) and broccoli and cauliflower (3.7–6.9 €/kg) are significantly less expensive or are at least affordable compared with the prices of commonly consumed animal food sources (chicken fillet (8.8 €/kg), eggs (1.7 €/10 eggs), milk (1.3 €/L), cheese (5–8 €/kg), salmon fillet (19 €/kg) and sardines (4.8 €/kg)) [320,321,322,323]. Nevertheless, for a better understanding of the affordability of foods which form the basis of a vegan diet, it is necessary to consider the level of self-sufficiency, both among individuals and nationwide, as well as the future goals of agricultural policy [324,325].

### 7.5. Specific Characteristics of a Vegan Diet/Lifestyle

When referring to a vegan lifestyle, all the structures/aspects of recommendations for a healthy and active lifestyle for the general population should be considered [43,44,45,326].

Numerous new challenges may arise with demanding changes in vegan dietary behavior. These challenges also include (i) social pressures (e.g., from family members, coworkers and friends who may not support individuals changing to a vegan diet), (ii) expected behaviors (e.g., associated with cultural norms), (iii) many logistical challenges (e.g., meal planning and food preparation), (iv) limited meal choices for a well-designed vegan diet when outside the home environment (e.g., balanced meals in a whole-food, plant-based diet) and (v) an urgent need to acquire new skills (e.g., organization and preparation of delicious balanced meals at home and away from home) with which to successfully change eating habits [18,98,174,327,328].

When individuals are introduced to a vegan diet, they often face challenges related to (1) different food tastes, (2) financial constraints and even (3) diet-specific issues, such as bloating or increased flatulence. Given these factors, predictors of an easier transition to a vegan diet and long-term dietary adherence may include an extensive support system, a positive social environment, convenience and food accessibility [57,98,327,328].

For an effective transition to a vegan diet, until such a diet is more widely accepted, the cooperation of the public health system and the support system service of the private sector is needed to ensure continuous integrated, branched and diverse vegan diet services.

### 7.6. Animal-Based Analogues/Alternatives

With the increased popularity of a vegan diet/lifestyle, alternatives to various animal-based products available (e.g., meat analogues, plant-based milk/beverages and vegan cheeses, to name just a few) [283,284,329,330,331], are more and more present in grocery shops (not only small-sized, specialized shops, but increasingly in commonly available supermarkets) in Slovenia.

In the last decade, vegan alternatives have expanded considerably [329,330]. However, caution is needed because they should not be automatically equated within their food group (i.e., coconut oil-based cheese vs. cashew nut-based cheese), nor should they be compared with animal-based food groups (i.e., coconut oil-based or tofu-based cheese vs. dairy-based cheese), both in terms of composition and contribution to nutritional adequacy.

Nevertheless, these vegan products are designed to resemble the taste, appearance and sensory characteristics of traditional animal-based products (i.e., burgers, sausages, milk and cheeses) [329,331]. In addition, these vegan alternatives to animal-based products can be consumed by people adhering to different dietary patterns, either (i) as an aid in the transition to a vegan diet, (ii) as a dietary goal to eating less animal-based foods or (iii) as a low-cost ingredient or food enriching an individual’s diet. Moreover, people on a well-designed vegan diet might not always look for specific vegan alternative products in order to mimic the nutritional profile of animal-based products from the same food group.

Importantly, consumers require products that are sustainable, palatable, safe, nutritious, available and affordable [330]. Therefore, not all animal-based food substitutes are healthy, and some of them are even ultra-processed, which can have various effects on health [330]. A recent randomized crossover control study compared the effect of consuming vegan alternative meat as opposed to animal meat on health factors among generally healthy adults. All other dietary (nonproduct) components were similar, and the researchers found no adverse effects on risk factors from the vegan products. However, the “plant phase” lowered the concentration of LDL cholesterol, probably due to lower saturated fat levels, a higher fiber content and plant protein in the vegan alternative meat [331]. In addition, Spanish researchers recently evaluated the nutritional composition of various plant-based cheeses (i.e., made from coconut oil, cashew nuts and tofu). Not surprisingly, coconut oil-based cheese (e.g., high in saturated fat and sodium) cannot be considered healthy [329], while cashew nut- and tofu-based cheeses both showed a healthier nutritional profile. The replacement of dairy-based cheese with cashew nut-, tofu- and plant-based alternatives could reduce intakes of salt and total fats, while replacing the intake of saturated with unsaturated fats [329].

In conclusion, the above-mentioned vegan alternatives to animal-based products have various advantages (i.e., high quality protein and more environmentally sustainable when soy is used) and disadvantages (i.e., some might have a high content of saturated fat, sodium, sugar and other (potentially artificial) sweeteners and flavors, or might have a higher environmental impact). Nowadays, a conscious consumer is faced with ever increasing challenges when selecting healthy vegan products as an alternative to animal ones [330]. At this point, there is still an open question about the long-term health impact of consuming some of these products (e.g., meat analogues, which are on one hand often ultra-processed, while on the other they contain high amounts of fiber and protein). Therefore, we only recommend that these products (except for products such as tofu-based cheese or plant-based milk/beverages without added fats and sugars) might not be consumed on a daily basis. In addition, they should contain as little as possible of saturated fat, sodium and sugar.

## 8. Conclusions

Given the critically high prevalence of common chronic diseases, the failure of different approaches to address these public health challenges and encouraging results regarding vegan diets in recent decades, we confidently contend that a well-designed vegan diet combined with a healthy and active lifestyle is a viable option for healthy adults who choose it. The evidence is clear, whereas the concerns are related to an inappropriately designed vegan diet, which is also a problem with any kind of diet (e.g., omnivorous).

A well-designed vegan diet should be energy and nutrient adequate and include proper supplementation, at least with vitamin B_12_, vitamin D in the autumn/winter months and possibly EPA/DHA. In addition, for physically healthy and active individuals (e.g., fitness enthusiasts and competitive vegan athletes) or as a compromise in the present fast-paced lifestyle, using plant-based protein supplements or meal replacements as well as sports drinks and creatine supplementation sparingly might be an option that has important benefits, for example, for body mass management, recovery after PA, adherence to a vegan diet or sport performance without compromising overall health.

In many aspects, but according to the totality of evidence, it is possible to argue that a well-designed vegan diet that is properly supervised by physicians (e.g., familiar with the vegan diet) should also be considered safe and healthy in pregnancy, infancy and childhood, especially in view of the current health challenges of this population and the general population as well due to the trend of current unhealthy diets and their evident consequences. However, more high-quality, long-term studies are needed to suggest further modifications to the vegan diet.

Importantly, we suggest that a well-designed vegan diet should be followed up by an expert in vegan diet and regularly medically supervised for possible nutrient deficiencies (i.e., vitamin B_12_, 25-hydroxyvitamin D, iron and iodine).

Given the above, we strongly believe that placing a well-designed vegan diet for Slovenes will also encourage media activities to regularly and objectively report the results of upcoming scientific studies of vegan diets, thereby indirectly contributing to the greater accessibility of well-designed vegan meals outside the domestic environment (e.g., at work, in educational institutions). In addition, better education, greater social responsibility and public–private sector participation are required for implementing well-designed vegan diets and lifestyles.

Regardless, future studies should include larger samples and long-term prospective randomized controlled studies where different but well-designed diets are compared (i.e., based primarily on unrefined dietary foods). In addition, there is also a lack of well-designed studies about physically active vegans and vegan athletes, as well as research that includes a vegan diet during pregnancy, breastfeeding, infancy, childhood and among older adults. Finally, we firmly encourage researchers in Slovenia to conduct further reliable scientific research in the field of the “vegan diet/lifestyle”.

## Figures and Tables

**Figure 1 nutrients-13-04545-f001:**
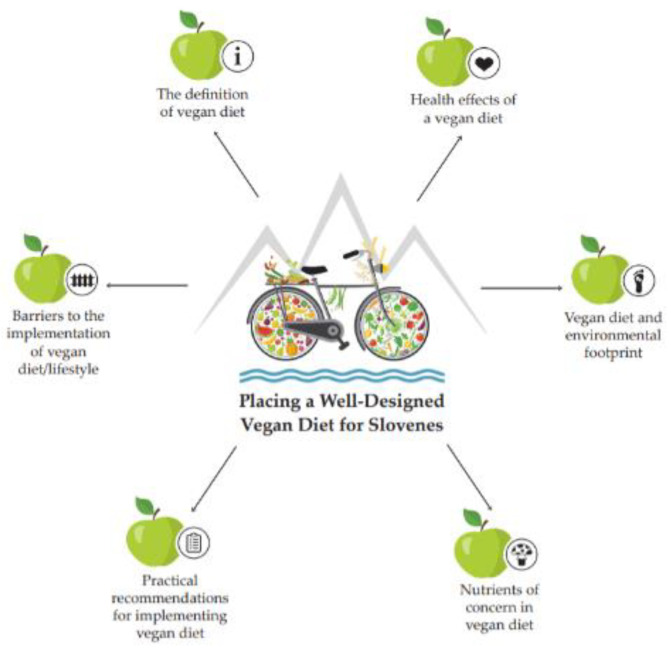
Placing a Well-Designed Vegan Diet for Slovenes.

## Data Availability

Not applicable.

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
