# Peer review of "Placing a Well-Designed Vegan Diet for Slovenes"

_nutrients, 2021, doi:10.3390/nu13124545_

Round 1

Reviewer 1 Report

Thank you very much for the opportunity to review this interesting manuscript. The manuscript covers a very important topic. The author is a renowned and well-known expert in the field of vegan diets and plant-based nutrition. Furthermore, the author and his team conducted multiple trials on vegan diets in the past. As such, he has considerable experience in that particular field.

The article is well-referenced, which is a big plus. The article will make an important contribution to our current understand of vegan diets. The fact that the author discusses his findings in the Slovenian context makes this manuscript special. I am very much looking forward to see this manuscript published. Prior to publication, however, I recommend a few revisions. Please find my detailed suggestions below:

Major revisions:

Please revise the English. The manuscript contains several very long sentences which are often difficult to read: e.g. page 2, lines 59 – 65; page 5, lines 222-229, page 8 lines 389 – 397, (…). These very long sentences are also hard to understand, and the reader will have difficulties to focus on the topic. Please kindly ask a native English-speaking friend or colleague to proofread the paper.

The manuscript would benefit from a graphical abstract or some kind of figure. The article is very long but contains neither tables nor figures. While not a must, I strongly encourage the author to change this (it will make the article more attractive for readers).

Minor revisions

Page 1, line 39: Please revise and add the word “diets”. I hope that nobody likes eating vegans.

Page 2, lines 55-56: This sentence requires a citation. Please kindly add a citation to support these claims.

Page 2, lines 64-65: You may wish to move this sentence to the conclusion section.

Page 2, lines 67-69: I suggest using Arabic numerals as you did in the text.

Page 2, lines 72-74: This sentence is unclear, please revise.

Page 2, lines 80-84: if possible, please add a reference.

Page 2, line 87: this is an excellent point! I suggest you revise this sentence in a way that there are many different definitions of a vegan (plant-based) diet, which makes comparison between studies and interventions. difficult

Page 2, line 94: I believe you mean “avoidance of prolonged sitting”. Please revise.

Page 3, line 105: What is the SI Menu? Please explain in detail.

Page 3, line 113: instead of “positive effects” I suggest “to reduce body mass and to improve body composition”.

Page 4, line 153: I suggest to use simple past (had instead of have)

Page 5, line 218: you may wish to add that adherence to a low-fat vegan diet was often higher than adherence to conventional diets in several studies. This is an important factor because “low adherence” is often used as a (false) argument against plant-based diets.

Page 7, line 349: While I agree, I suggest to replace shocking with “concerning”.

Page 8, line 364: This sentence is grammatically wrong.

Page 9, line 430: This sentence is one of several complex sentences in the manuscript. Please revise and try a simpler English language, e.g. “There are several nutrients of particular concern in a vegan diet”.

Page 9, line 456: This is correct. Please add that these references refer to a “low-fat” vegan diet, a specific form of the vegan diet with a low fat content (and thus a reduced energy density).

Page 10, line 509: It is controversial whether and how this score (Digestible Indispensable Amino Acid Score (DIAAS) that was recently suggest to replace the PDCAAS) should be used with a plant-based diet (please see Craddock et al. 2021, Current Nutrition Reports).

Page 11, line 549: insert the word “common”; e.g. “and common among people of all ages”.

Page 12, line 616: can be easily met with what? Please complete the sentence

Page 13, line 646: “was close to 30% of energy intake …”; please add the word total.

Page 14, line 674: replace “does” with “did”.

Again, I would like to thank you for the opportunity to review this manuscript. I am sure this will make a great contribution. I believe the manuscript will benefit from my suggested corrections.

I commend the author on his excellent work!

I wish you all the best, please stay safe from the COVID-19 pandemic!

Reviewer 2 Report

The author brings a relevant review about the vegan diet focused on the Slovenes, summarizing the impact of a vegan diet on human health, as well as nutrients that vegans might be more susceptible of deficiencies and limitations of following a vegan diet. It is quite an extensive review but very easy to read, in good English, and provides a robust source of information about vegan diets that can be useful for the health professionals in Slovenia but also worldwide. 

I have a few comments and suggestions for improvement:

Line 39: I would say there is an interest in following a vegan diet or eating a vegan diet instead of "There is an increasing interest in eating vegans." 

Moreover, why there is an increase in interest in following a vegan diet? Is this supported by research? Or there is more interest in reducing meat consumption, rather than cutting completely meat from the diet? I missed supporting information for this statement and there are studies done for instance from ProVeg or The Good Food Institute about this.

Line 39: 3% of how much? How many people participated in the survey?

Line 85: "... meals prepared with healthy methods of preparation..." What does mean healthy methods? Give examples. "... meals eaten to satiety..." Meals that induce satiety or eating food that promotes satiety or promoting satiety food would be better here. 

Line 215: What is the mechanism behind the vegan diet improving beta-cell function and insulin sensitivity? I think this could be a bit more elaborated. Is it just the diet or a combination with lifestyle? Was there any association with the type of food or any specific nutrient?

Line 291:  I found in the literature some papers that evaluated vegan athletes. I suggest the following reading and evaluation of including these references in the paper. I also suggest another literature search about this topic:

Wirnitzer, K., Seyfart, T., Leitzmann, C. et al. Prevalence in running events and running performance of endurance runners following a vegetarian or vegan diet compared to non-vegetarian endurance runners: the NURMI Study. SpringerPlus 5, 458 (2016). https://doi.org/10.1186/s40064-016-2126-4

Lynch, H.; Johnston, C.; Wharton, C. Plant-Based Diets: Considerations for Environmental Impact, Protein Quality, and Exercise Performance. Nutrients 201810, 1841. https://doi.org/10.3390/nu10121841

Davey, D.; Malone, S.; Egan, B. Case Study: Transition to a Vegan Diet in an Elite Male Gaelic Football Player. Sports 20219, 6. https://doi.org/10.3390/sports9010006

Wirnitzer, K.; Boldt, P.; Lechleitner, C.; Wirnitzer, G.; Leitzmann, C.; Rosemann, T.; Knechtle, B. Health Status of Female and Male Vegetarian and Vegan Endurance Runners Compared to Omnivores—Results from the NURMI Study (Step 2). Nutrients 201911, 29. https://doi.org/10.3390/nu11010029

Line 311:  it was already mentioned “when breastfeeding is not possible” earlier in the same sentence.

Line 374 What was the percentage of iron deficiency among non-omnivores? It is interesting to show that this deficiency and vitamin B12 deficiency can also happen in a non-vegan or vegetarian diet.  Were these children being guided by a nutritionist or physician? I was wondering why in this study children had deficiencies while in the Finland study children were eating 51% more iron than non-vegan children.   

Section 4: It might be relevant to comment about plant-based products, such as meat analogs, that depending on their production the environmental footprint is not that low.

Line 484: 0.8 and 1.0 g?

Line 525: Why is food preparation between “ “ ?

Line 660: I guess one of the problems to obtain micronutrients from plant-based sources, besides antinutritional factors is the need to eat large quantities compared to for instance drinking a glass of milk. I think this should be addressed. And this could be one of the limitations for vegans to properly intake calcium.

Line 668: why cooking is written between “ “?

Line 755 and 760:  if commercial plant-based milk is chosen advise that it is preferable to choose products fortified with calcium and Vitamin D and B12, when possible.

With the increase in plant-based products in the market (e.g. meat analogs, plant-based milk, and cheese), I believe a section about these products should be included. Are there studies/reports showing the consumption of these products by vegans? What is the nutritional quality of these products? Can vegans use these products to replace animal-based products in terms of nutrient composition, such as protein? Can vegans include these products in their diet and still have a healthy diet?  think a recommendation about the products should be given, considering their nutritional content. Information about these products will also help health professionals advise their patients. 

Line 911 which skills?

In terms of financial constraints, nothing was mentioned about the prices of plant-based products. Are there studies about this? The price of these products is still higher than the animal-origin ones. Is this limiting their consumption or vegans would consume anyway? How is this in the context of the country? I believe there are some reports from GFI and or ProVeg that analyzed this.

Conclusion section: Which kind of research is needed in the field of a vegan diet? What is actually missing?

Reviewer 3 Report

This very comprehensive review provides an evidence-based oversight on the advantages of the vegan diet/lifestyle. The paper is well written, although the author obviously is a "superfan" of veganism. I have the following remarks:

  • On average, vegans are required to eat about 30% more than omnivores on a daily basis, in order to meet the necessary proteinic and caloric intake. This, in turn, augments the costs of the vegan lifestyle, and the author should make a note of these remarks in section 5 or 6.
  • While proteins are well and extensively discussed, there also exist amino acids that are not incorporated in proteins. Typical ones are carnosine and taurine, which are mostly found in meat and appear to yield antioxidant effects. The vegan diet mostly excludes these possibly protective amino acids (e.g. cancer), and there should be a note of this in section 4.
  • When micronutrients are discussed, Selenium appears to missing, and should be summarized briefly in section 4.
  • The vegan lifestyle might protect against (some forms of) dementia, which could be briefly mentioned in section 2.

Round 2

Reviewer 1 Report

The author did an excellent job and substantially improved the paper. Figure 1 and the English language revisions are highly appreciated. Thank you! Some minor spelling / grammar mistakes remain and will require a final proofreading. I recommend publication of this review. Well done!